# Validation of Contact Mats to Monitor Dairy Cow Contact with Stall Partitions

**DOI:** 10.3390/ani10060999

**Published:** 2020-06-08

**Authors:** Caroline Freinberg, Athena Zambelis, Elsa Vasseur

**Affiliations:** 1Department of Animal Science, McGill University, Ste-Anne-de-Bellevue, QC H9X 3V9, Canada; crf2137@columbia.edu (C.F.); athenaczambelis@gmail.com (A.Z.); 2Environmental Biology, Columbia University, New York, NY 10027, USA

**Keywords:** animal movement, contact mat, dairy cow, stall comfort

## Abstract

**Simple Summary:**

In indoor housing systems, repetitive contact between dairy cows and stall partitions may reduce their quality of rest and hinder ease of movement. A contact mat (CM) is a metal band that produces an electrical signal in response to a contact force. The objective of this study was to validate the ability of a CM system to monitor cow contact with stall dividers and neck rail when compared to visual observation. Eleven lactating cows were monitored both visually and with the CM system for 4 h/d for 4 consecutive days in a tie-stall housing system. Individual CM were affixed to the stall dividers and neck rail to record the frequency of cow contact per second based on a minimum of 11.3 kg of pressure required to produce a digital output. Two trained observers recorded the frequency of cow contact against the stall partitions on a per second basis using predefined criteria. The results suggest that the CM system can be used to accurately rank cows based on their frequency of contact with the stall dividers for both scientific and management purposes. The CM system requires modification and validation if to be used to record the exact number of contacts made by cows with the stall partitions.

**Abstract:**

In indoor housing systems, repetitive contact with the stall partitions may reflect issues between the housing environment and the cow, by reducing the quality of rest and hindering ease of movement. The objective of this study was to validate the ability of a contact mat (CM) system to monitor cow contact with stall dividers and neck rail when compared to video observation. Eleven lactating cows were monitored using video recording and with the CM system for 4 h/d for 4 consecutive days. CM were affixed to the stall dividers and neck rail to record the frequency of cow contact per second. Two observers recorded the frequency of cow contact against the stall partitions per second using three criteria: cow contact with the stall dividers or neck rail regardless of CM contact (Total Rail Contact), cow contact with the CM regardless of placement or force (Total CM Contact), and cow contact with the CM through proper placement and force (Effective CM Contact). The correlation strength used to measure agreement between video observation and CM system for cow ranking based on degree of contact varied from moderate to very high (dividers: Total Rail Contact: r_s_ = 0.68, Total CM Contact r_s_ = 0.90, Effective CM Contact r_s_ = 0.90; neck rail: Total Rail Contact: r_s_ = 0.71, Total CM Contact r_s_ = 0.66, Effective CM Contact r_s_ = 0.58). CM can be used to accurately rank cows based on their frequency of contact with the stall dividers. This can be used to identify individual cows that need intervention for stall comfort risk factors, or to assess which indoor housing environments offer fewer constraints and better movement opportunities.

## 1. Introduction

With the transition of dairy cow husbandry to more intensive systems, the welfare concerns associated with indoor housing have become a major research focus. Studies have identified the frequency of collisions with housing equipment while lying down as a welfare problem and have included it in on-farm assessment protocols as an outcome measure [1,2]. However, little literature exists on the actual frequency of contact between dairy cows and the confines of their housing environment. Repetitive contact with the stall partitions may reflect problems between the housing environment and the cow, by reducing the quality of rest and hindering ease of movement [2]. Stall configurations that impede movement have been shown to impact the prevalence of injuries in dairy cows [3,4,5]. For example, tie-stall cows with neck rails of 116 to 132 cm were significantly correlated with a 70% decrease in the prevalence of neck lesions compared to cows with neck rails of lower heights [5]. Assessing the frequency of cow contact with stall rails could help determine which stall configurations are optimal for cow comfort.

Visual assessments of farm animals are costly and time-intensive, as observers need to be properly trained and sufficient data needs to be visually collected [6]. In order make the data collection process more time-efficient, automated methods can be used to monitor risk factors for poor welfare. Contact mats (CM) are metal bands that detect pressure in response to a contact force and then generate an electrical signal. Other types of pressure-sensing mats have been used previously in studies on animal behavior, though these mats are commonly placed on the ground to evaluate gait [7]. These include the GAITFour pressure mat gait analysis walkway system and static force plates, which can measure weight distribution on the limbs of sows and other species when standing and walking to detect lameness [8,9]. Developing CM as a method to autonomously measure cow contact against stall rails could be a useful tool to help improve welfare, by identifying problematic stall configurations or individual cows that require housing intervention. The objective of this study was to validate the ability of a CM system to monitor cow contact with stall dividers and neck rail when compared to video observation.

## 2. Materials and Methods

In April of 2018, primiparous (*n* = 5) and multiparous (*n* = 11) lactating Holstein cows were randomly selected from the McGill University Macdonald Campus Dairy Complex (Ste-Anne-de-Bellevue, QC, Canada) for study enrolment. All experiments were designed in compliance with the institutional Animal Care Protocol 2016-7794, and involved the use of non-invasive techniques, principally the use of video recordings for behavioral observation. The trial period lasted 8 weeks, with 16 cows being video recorded in total. The final dataset included 11 of the 16 cows (primiparous: *n* = 4, multiparous: *n* = 7; 170.2 ± 60. DIM), since 5 cows had to be excluded from the study due to technological malfunctions or inadequate camera views due to obstruction. A sample size of 11 cows was deemed sufficient to statistically evaluate the agreement between CM versus video observation, based on a correlation analysis with 0.90 power, an α of 0.05, and an assumed effect size of at least 0.75 [10]. 

Cows were housed in a single row of tie-stalls that had dimensions following current recommendations made by the Dairy Code of Practice based on measured body dimensions of the cows (average hook bone width: 65.8 ± 3.8 cm, average height: 153.4 ± 2.5 cm) [11], and remained in the same stall throughout the data collection period. The average stall width was 141.0 ± 6.9 cm (mean ± SD), the average stall length was 185.2 ± 5.3 cm, and the average neck rail height was 118.1 ± 8.4 cm. The stall bases consisted of rubber mats (KKM longline; Distribution Multi-Mat, Inc., Ste-Cécile-de-Milton, QC, Canada) bedded with either straw or wood shavings. Straw was supplemented twice a day at 7:00 a.m. and 4:00 p.m. to maintain at least 7.6 cm of bedding depth. Wood shavings were supplemented once each morning to maintain 2 cm or less of bedding depth. The stalls and gutters were scraped regularly by barn staff from 5:00 a.m. until 9:00 p.m. Milking was carried out in-stall twice daily at a 12 h interval. Cows were fed a TMR four times daily at 6:00 a.m., 10:00 a.m., 4:00 p.m., and 7:00 p.m., with feed pushed up 6 times per day to ensure food availability. Water was accessible ad libitum from a self-filling water bowl that was shared between each pair of stalls. 

In the month prior to the start of the trial period, each stall was equipped with 4 individually sealed CM (925 Series 25 ft × 30 in Pressure Mat Roll, United Security Products, Poway, CA, USA). The CM roll was cut length-wise into individual 5.1 cm × 76.2 cm CM for each of the dividers and a joint pair of CM for the neck rail (total of 4 CM per stall). The metal ends of each CM were then revealed from the water-resistant plastic coating and soldered to 1.5 m of 18 American Wire Gauge (AWG) wire (Figure 1a). The length of the 18 AWG wire and the soldered ends of each CM were coated with electrical tape to prevent electrical current from escaping the circuit. CM were adhered to the stall rails using duct tape (Cantech, Montreal, QC, Canada) at each end and clear packing tape along the length of the metal (Figure 1b). One CM was placed on the bottom rail of each divider, centered at approximately a 45° angle to the ground (CM2, CM3 in Figure 2). A joint pair of CM were centered on the inside of the neck rail, with the top CM parallel to the rail and the bottom CM at an approximate 45° angle to the ground (CM1 in Figure 2).

At the start of each week, CM were coated with an anti-cribbing paste (Cribox Anti-Cribbing Paste, Hydrophane, Lincoln, UK) to avoid damage from chewing. The 18 AWG wires from each CM were affixed to the stall rails and routed to a separate area of the barn, where output data was collected. Each 18 AWG wire was connected to a programmable logic controller (PLC; CLICK Ethernet Analog PLC, AutomationDirect.com, Georgia, GA, USA), which produced an electrical signal and assigned a timestamp when a force of at least 72.9 kg of pressure per square cm was applied to a CM in each stall. For context, an equivalent force pressure was measured by a penetrometer (FT 327 Penetrometer, QA Supplies LLC, Virginia, VA, USA) when flipping a standard light switch on or off. A computer was connected to the PLC via an ethernet cable, and the PLC was programmed using the CLICK Programming Software (Version 2.00, AutomationDirect.com, Atlanta, GA, USA). The PLC was connected via an 18 AWG wire to a touch-screen display board and data storage system (C-more EA9 series Touch Screen, AutomationDirect.com). A computer was connected to the touch-screen display board via an ethernet cable, and the data storage system was programmed to store the electrical signals produced by the PLC as data using the C-more Windows-based programming software (Version 6.20, AutomationDirect.com). Using this system, the number of contacts registered by the CM circuit were stored on a continuous basis for all cows throughout the study period. The presence or absence of contacts was displayed on a per second basis with a unique identifier for each CM in each stall. A USB was connected to the display board at the end of each day to collect and store the data for the previous 24 h. CM were manually tested on the first and last days of the trial period to ensure circuit functionality, by applying pressure to the CM by hand and confirming the recording of an electrical data signal.

Video recordings were used to assess agreement between visual observation versus CM for the frequency of stall rail contact during pre-selected times. Cows were video recorded 4 h/d for 4 consecutive days (a total of 176 h). Video recordings were taken continuously for four 1 h time blocks, scheduled from 10:45 a.m. to 11:45 a.m., 12:00 p.m. to 1:00 p.m., 2:30 p.m. to 3:30 p.m., and 3:45 p.m. to 4:45 p.m. The duration of each block was selected to be 1 h since that was within the maximum battery life of the video cameras to prevent data loss. The timing of the blocks was chosen to avoid periods of high disturbance to the cows or high human activity in the barn, specifically for in-stall milking or concurrent experiments being run. To obtain an overhead view of cow positioning within each tie-stall, a ceiling video camera was used (HERO4 Silver, GoPro, San Mateo, CA, USA or Smart Turret 2.8, Hikvision Digital Technology Co., Ltd., Hangzhou, China; 720p at 60 frames per s; height: 338 cm centered above each stall; C4 in Figure 2). Two video cameras (HERO4 Silver, GoPro; 720p at 60 frames per s) were positioned to obtain a lateral view of the underside of each divider, and were affixed to the bottom of the divider (27 cm from the front of the stall; C2, C3 in Figure 2). One video camera (HERO4 Silver, GoPro; 720p at 60 frames per s) was positioned to obtain a side angle of the inside view of the neck rail, and was adhered to a horizontal support bar affixed to the neck rail at the side of the stall (16 cm from the front of the stall; C1 in Figure 2). At the start of each block, the session time was calibrated for each view by displaying the local time to each video camera using the website https://time.is/. VSDC Video Editor Pro (version 5.8, Flash-Integro LLC) was used to compile the 4 different video camera views onto a single screen to facilitate scoring, using the displayed time at the start of each block as a common timepoint.

Each block of video recording was scored by two trained observers, who recorded the frequency of cow contact against the stall partitions on a per second basis using 3 criteria: cow contact with the stall dividers or neck rail regardless of CM contact (Total Rail Contact), cow contact with the CM regardless of the placement or force used (Total CM Contact), and cow contact with the CM through proper placement and force to create an output signal (Effective CM Contact). An instance of contact was defined as occurring when the cow visibly and obviously collided with or contacted the stall partitions with any part of the body. In addition to this, the force of contact with the stall was recorded as either: soft if the contact had a light impact and caused no rail recoiling, or hard if the contact had a forceful impact that caused rail recoiling. Instances of contact lasting a maximum of 4 s were recorded as a bump, while instances of contact lasting longer than 4 s were classified as a lean. Lastly, the stall partition involved in the contact was recorded as either the left divider, right divider, or neck rail. To assess the consistency of scoring across observers and time respectively, inter- and intra-observer repeatability were tested at the start of the trial and again after scoring each cow. Inter- and intra-observer repeatability were evaluated by comparing the presence or absence of contacts per second using all 3 scoring criteria, based on 18 1 min clips of 3 cows. All the categories had almost perfect agreement, with an average Cohen’s kappa coefficient (±SD) of 0.91 ± 0.06 for intra-observer repeatability, and 0.89 ± 0.07 for inter-observer repeatability (following the system of Landis and Koch, 1977) [12].

The raw data collected from video observation included presence (1) or absence (0) of stall contact on a per second basis for each cow for each stall partition, in addition to contact force, cow behavior at the time of contact, and contact duration (bump or lean). In contrast, however, the raw data from the CM system consisted only of presence (1) or absence (0) of contact on a per second basis for each cow for each stall partition. All additional information collected during video observation was not compared statistically to the CM system data, due to a lack of comparable information provided by the CM system. The additional measures collected through video observation have only been used to interpret differences in agreement against the CM system. Comparing agreement between the CM system with the 3 scoring categories of video observation (Total Rail Contact, Total CM Contact, Effective CM Contact) was used to determine whether the CM system can differentially detect contact depending on variations in force and placement.

First, the distribution of the data was assessed to check for normality. The distribution of the CM data was not found to be normally distributed. However, the stall contact data collected through video observation was confirmed for normality. It is likely that contact with stall partitions is a normally distributed variable. However, the low sensitivity of the CM system produced a right-skewed non-normal distribution, where the number of contacts for each cow tended to favor 0. No transformations were applied to the data and all statistical analyses chosen were non-parametric in nature. Due to the repeated measures design of the study and the non-parametric nature of the statistical analysis, agreement between the CM system and video observation could only be assessed using a correlational analysis of the ranking of each cow for degree of stall contact. No additional cow-level or behavioral variables were included in the analysis directly but were used in the discussion to interpret the results. The type of contact in terms of force and placement was considered for agreement with the CM system using the three categories of video observation: Total Rail Contact, Total CM Contact, and Effective CM Contact. A Wilcoxon Signed-Rank test was used to compare video observation and CM system agreement on the total number of contacts made by each cow over the entire trial period with the dividers or neck rail. A Spearman’s correlation was used to compare video observation and CM system agreement on the ranked position of each cow based on the average number of contacts per min with the dividers or neck rail. The correlation coefficients were categorized according to the criteria of Hinkle et al. (2003) as negligible (0.00–0.30), low (0.30–0.50), moderate (0.50–0.70), high (0.70–0.90), or very high (0.90–1.00) [13].

## 3. Results

The Spearman’s correlation coefficients for measuring video observation and CM system agreement on the ranked position of each cow based on the average number of contacts per min with the dividers or neck rail are presented in Table 1. The correlation between the CM system and video observation for the average number of contacts per min with the dividers was moderate (r = 0.68; *p* = 0.18) for rail contact, very high (r = 0.90; *p* = 0.0002) for CM Contact, and very high (r = 0.90; *p* = 0.0002) for Effective CM Contact. In contrast, the correlation between the CM system and video observation for the average number of contacts per min with the neck rail was high (r = 0.71; *p* = 0.01) for Rail Contact, moderate (r = 0.66; *p* = 0.03) for CM Contact, and moderate (r = 0.58; *p* = 0.06) for Effective CM Contact.

The mean, standard error of the mean (SEM), and range for the total number of contacts made by each cow with the dividers or neck rail, as well as the analysis of CM system versus video observation agreement, are also presented in Table 1. The average total number of contacts made by each cow with the dividers was found to be consistently underestimated by the CM system compared to video observation, with Rail Contact being 91.3% lower (*p* = 0.001), CM Contact being 86.4% lower (*p* = 0.001), and Effective CM Contact being 86.2% lower (*p* = 0.002). For the average total number of contacts made by each cow with the neck rail, this underestimation was found to be further amplified, with Rail Contact being 98.7% lower (*p* = 0.001), CM Contact being 97.2% lower (*p* = 0.001), and Effective CM Contact being 96.2% lower (*p* = 0.001) when recorded by the CM system versus video observation. The average total number of contacts made by each cow was consistently higher for the dividers compared to the neck rail, with 451.7% more Rail Contact, 615.7% more CM Contact, 816.7% more Effective CM Contact, and 2951.2% more contacts registered by the CM System (Table 1).

## 4. Discussion

Based on the results, the CM system can be used to rank cows from highest degree of contact to lowest degree of contact with the stall dividers. Based on this evidence, the CM system demonstrated a strong ability to rank cows by their degree of contact with the dividers, but only for contact in the area where the CM was placed and regardless of force. However, the CM system did not demonstrate a strong ability to rank cows by their degree of contact with the neck rail, regardless of whether the entire rail was considered or only the area where the CM was placed. In terms of application, the CM system could be used to compare different stall configurations by ranking cows based on their degree of divider contact. The extent of divider contact may act as a risk factor for dairy cow welfare across stall configurations, based on possible associations with cow ease of movement and susceptibility to injury. Furthermore, the CM system could be used from a management perspective to identify cows within a herd that rank highly for divider contact and may require intervention for stall comfort risk factors. To determine the feasibility of implementing a CM system for future management or research use, key components of cost, labour, and time would need to be considered. The cost of the CM system was approximately $50.33 USD per tie-stall for all materials used, not including expenses related to labour. A cost estimate has not been provided for labour, due to variation in factors such as expertise, regionality, and wage. However, installation of the CM system for 12 stalls required approximately 35 h of labour for a group of 5 workers, or 175 h total (equivalent to 14.6 h per stall). Fortunately, the system required minimal maintenance for the duration of the trial period once installed, and damage to specific CM units can be replaced without a need to reinstall the entire system. While the CM system has been validated for use in a tie-stall housing system, its use in free-stalls for monitoring of individual cows would require the addition of an electronic identification system, since different cows make use of a single stall. Similar recognition systems have been used to monitor feeding behavior in electronic feed and water bins, with the use of an ear tag containing a unique passive transponder for each cow [14]. Any modification to the CM system to adapt it to an alternative housing system or improve its accuracy would necessitate additional validation.

Inadequate placement of the CM on the stall partitions is the most obvious justification for why a discrepancy existed in the ability of the CM system to rank cows based on divider versus neck rail contact. It is possible that by using a joint CM for the neck rail or selecting an inappropriate angle at which to secure the CM, the area of the mat responsible for producing a signal was not in the necessary location. It is also possible that the nature of the contact applied at the neck rail compared to the dividers altered the ability of the CM system to accurately sense the pressure. Contact at the dividers consisted of 97% leaning and only 3% bumping, where the contact was more forceful since it involved the body weight of the cow lying down. In contrast, contact at the neck rail consisted of 72% leaning and 38% bumping, where the force of contact was much softer and involved brief brushes of the neck primarily during feeding. While difficulties with scoring can limit the ability of a study to properly validate technology, it is unlikely that this is the source of the discrepancy caused here. The three separate scoring criteria were created to increase the specificity with which different types of contact could be classified. The high inter- and intra-observer repeatability indicate that data collected through video observation was scored consistently.

While it is expected that there be an underestimation of Rail Contact by the CM system due to the absence of CM on all surfaces of the stall partitions, the strong underestimation of CM Contact and Effective CM Contact is indicative of a low signal production sensitivity. The low sensitivity of the CM system has been depicted in Figure 3, via a comparison of the CM system to video observation for the average number of contacts per min made by each cow with the dividers or neck rail. The CM system demonstrated a consistent underestimation of cow contact with stall partitions, with no instance of an overestimation of cow contact occurring. Since the magnitude of this underestimation of contact ranged from 0.2% to 99% between cows, there is no way to correct for it while maintaining accuracy. Since the correlation between the CM system and video observation was very high in terms of divider contact ranking, an inference can be made that the degree of underestimation of contact for each cow by the CM system was proportional to their degree of contact. In other words, cows that had lower underestimation ranges closer to the minimum of 0.2% were often the cows with less contact overall, while cows with higher underestimation ranges closer to the maximum of 99% were often the cows with more contact overall. As such, the CM system can only provide a baseline for the degree of contact occurring between individual cows and stall partitions, and not an exact number. Considering that the CM used in this study were designed for human gait analysis and security, any attempts to increase the sensitivity of the CM system for its application to dairy cows should involve modifications to the CM itself. Because the surface area implicated in cow contact is much larger than that of a human foot, the surface area of the CM that registers force may need to be increased. By increasing the distance between the foam separators that prevent the CM from producing a signal, the surface area that registers contact would increase and the force of contact required to produce a signal would decrease. We postulate that these changes would increase the sensitivity of the CM system to the pressure applied by cow contact.

Though some studies have identified that collisions with the stall partitions during lying down are a risk factor for dairy cow welfare, this is the first study that has quantified this type of contact over a sizable time period. Based on video observation, the total number of contacts with all stall partitions for a single cow ranged from a maximum of 23,814 (dividers: 18,024, neck rail: 5790) to a minimum of 3231 (dividers: 3064, neck rail: 157) over the 16 h observation period (Rail Contact; Table 1). This represents a maximum frequency of 24.8 contacts per min (23,814/960 min in 16 h) and a minimum frequency of 3.4 contacts per min (3231/960 min in 16 h) across cows, as verified by video observation. Due to the lack of previous literature to compare these values to, it is important to consider the data with caution. Even if the CM system data is considered instead as a baseline, the total number of contacts with all stall partitions for a single cow ranged from a maximum of 3663 (dividers: 3500, neck rail: 163) to a minimum of 39 (dividers: 39, neck rail: 0) over the 16 h observation period (CM System; Table 1). These values represent a strikingly high number of contacts with the housing system, considering that the data collection period is brief relative to the 5.5 year average lifespan of a dairy cow [15]. It would be interesting for future research to test whether the frequency of cow contact with the stall partitions acts as a dairy cow welfare risk factor, through an association with injury prevalence and other outcome measures for cow comfort. As a risk factor, frequency of stall contact could be used to assess how specific housing systems and stall configurations compare in terms of dairy cow welfare.

Since the discrepancy between dividers and neck rail contact was present for both the CM system and video observation, it is apparent that most collisions occur while the dairy cow is lying down, thus implicating the dividers only. While injury risk during lying-down transitions has been associated with hard incompressible lying surfaces [16], forceful collisions with the dividers may have a role in these findings. Because lying-down movements are pre-determined by skeletal and muscle structure, the successful adaptation of these motions to stall design is improbable [17]. The large SEM reported for the average total number of contacts with the stall dividers and neck rail indicates that there was a high degree of variability between individual cows (Table 1). Since all tie-stalls had dimensions following current recommendations made by the Dairy Code of Practice based on individual cow size, it is unlikely that space allotment was the source of this variation in the data [11]. Instead, it is likely that cow-level factors such as age, temperament, or size played a role in the extent of contact with stall partitions. It is also possible that frequency of contact depended on differences in how individual cows use the space that they are provided. Gaining an understanding of how cows use the space within their stalls may be a useful area for future research investigation that could lead to an improvement in the housing environments of dairy cows.

## 5. Conclusions

We believe this to be the first validation study on the ability of a CM system to monitor cow contact with stall partitions in an automated manner. The results suggest that the CM system can be used to accurately rank cows based on their frequency of contact with the stall dividers for both scientific and management purposes. The results also suggest that the CM system requires modification and further validation if to be used to record the exact number of contacts made by cows with the stall partitions. The CM system can be used to identify individual cows that may require intervention for stall comfort risk factors, as well as to assess whether different indoor housing environments offer fewer constraints and therefore better movement opportunities for dairy cows. A generalization of this assessment tool could be made to other species of animals kept in confinement.

## Figures and Tables

**Figure 1 animals-10-00999-f001:**
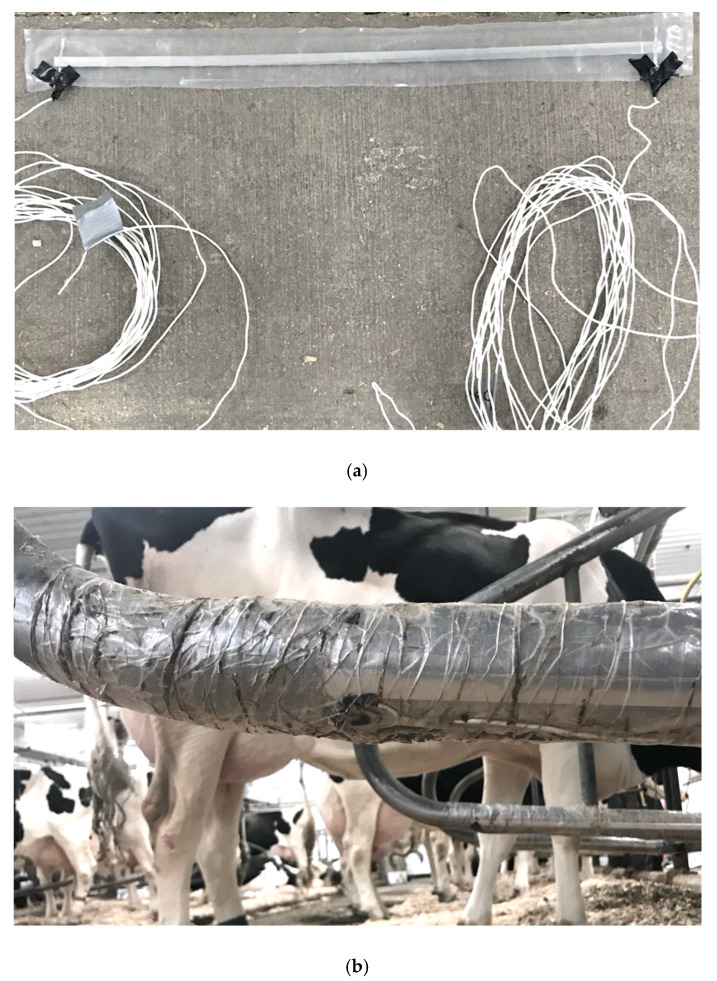
Photographs depicting (**a**) the full length of an individual contact mat (CM) within a water-resistant plastic coating, with the metal ends soldered to 1.5 m of 18 AWG wire and coated in electrical tape for insulation prior to stall attachment; and (**b**) a portion of an individual CM affixed to the left stall divider using duct tape (Canadian Technical Tape Ltd., Saint-Laurent, QC, Canada) at the end and clear packing tape along the length of the metal band. The CM shown has been positioned so that the pressure-sensing metal band is levelled at approximately a 45° angle to the ground. This photograph was captured at the end of the trial period to show the condition of the CM after an 8 week period.

**Figure 2 animals-10-00999-f002:**
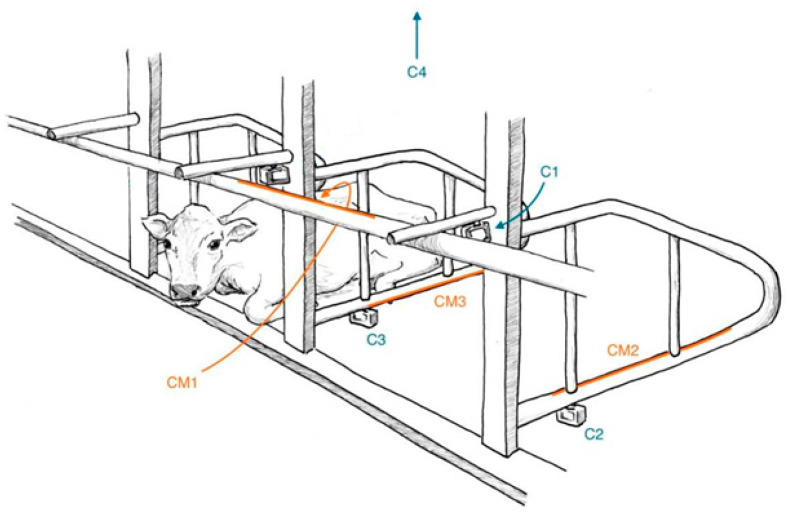
Illustration depicting the video camera and CM placement for a single tie-stall in accordance with study design. Three separate CM (orange) were used for each stall, with CM1 affixed to the neck rail, CM2 to the right divider, and CM3 to the left divider. Three video cameras (blue) were used to provide visual observation of each CM, with C1 used for the neck rail, C2 for the right divider, and C3 for the left divider. An additional video camera (C4) was included to provide an overhead view of each stall for monitoring cow position.

**Figure 3 animals-10-00999-f003:**
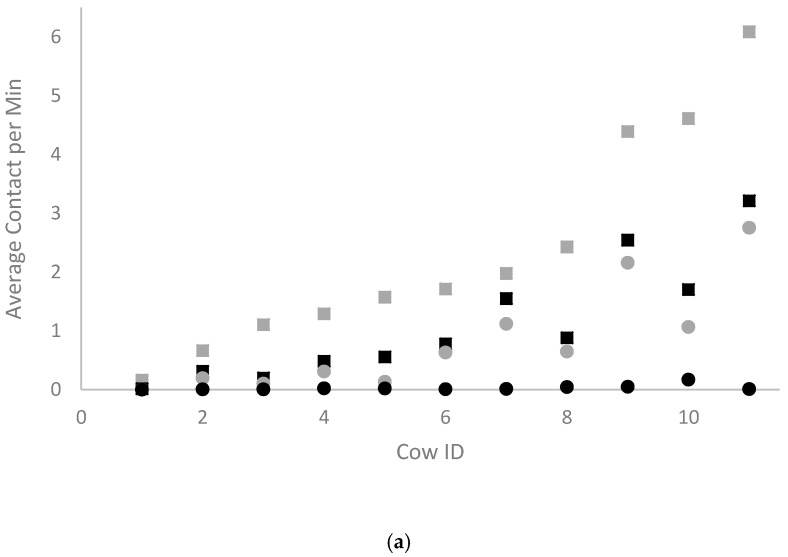
Plot of the average number of contacts per minute made by each cow (*n* = 11) with (**a**) the neck rail, or (**b**) dividers as recorded by the CM system (black circles) compared to video observation (Rail Contact: grey squares; CM Contact: black squares; Effective CM Contact: grey circles).

**Table 1 animals-10-00999-t001:** Average total number of contacts made by each cow (*n* = 11) over the 16 h observation period with the dividers or neck rail for the CM system versus video observation with *p*-values for the Wilcoxon Signed-Rank test, and Spearman’s correlation coefficient (r_s_) with *p*-values for analyzing the agreement between the CM system and video observation on the ranked position of each cow based on the average number of contacts per min with the dividers or neck rail.

Variable	Mean	SEM	Max	Min	*p*-Value for S	r_s_	*p*-Value for r_s_
Dividers Only							
Video Observation							
Rail Contact	10,132.1	1532.0	18,024	3064	0.001	0.68	0.02
CM Contact	6479.9	1378.4	11,715	113	0.001	0.90	0.0002
Effective CM Contact	6382.2	1376.7	11,477	41	0.002	0.90	0.0002
CM System	882.4	322.3	3500	39	-	-	-
Neck rail Only							
Video Observation							
Rail Contact	2243.0	535.2	5790	157	0.001	0.71	0.01
CM Contact	1052.4	293.3	3055	18	0.001	0.66	0.03
Effective CM Contact	781.5	255.6	2617	3	0.001	0.58	0.06
CM System	29.9	14.1	163	0	-	-	-

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
