# Peer review of "Validation of Contact Mats to Monitor Dairy Cow Contact with Stall Partitions"

_animals, 2020, doi:10.3390/ani10060999_

Round 1
Reviewer 1 Report
This manuscript was well written. This was really a pilot study. It would have been nice to not have some technical difficulties and a greater number of animals in the study. However, this is the first study utilizing this system with dairy cattle and the study did provide good information. The illustration was appropriate but some close up pictures of the system would make the paper better.
Author Response
Comments and Suggestions for Authors
Reviewer 1:
R1: This manuscript was well written. This was really a pilot study. It would have been nice to not have some technical difficulties and a greater number of animals in the study. However, this is the first study utilizing this system with dairy cattle and the study did provide good information. The illustration was appropriate, but some close-up pictures of the system would make the paper better.
AU: Thank you for your feedback. Given the novelty of the system, there were unforeseen technical issues that arose during the validation process, however all impacted samples were removed from the study to avoid having large missing data percentages that could bias the statistical analysis (L76-77). Fortunately, a sample size of 11 cows was found to be sufficient for a correlation analysis with 0.90 power and an α of 0.05 (L78-80). Due to constraints with equipment, we could not facilitate a larger sample. Two additional pictures have been added to better illustrate the study methodology (Figure 1a, b; L113-123), specifically the structure and stall attachment of a single pressure-sensing unit. Figure 2 (L166-172) depicts all areas of the stall where CM were placed, for which specific angles can be found within the methodology section (L109-112).

Reviewer 2 Report
This is a straightforward but important paper, as it details a means for automatically assessing the welfare of dairy cows. This is also potentially applicable to other settings, where a mechanized, automatic assessment of welfare could be extremely useful.
The difficulties, as the authors note, is that the ability to accurately assess some of the contacts were only moderately correlated. While the authors spend some time in their discussion suggesting ways to improve these automatic assessments, it still remains at least a partial issue.
Regardless, if the choice between some form of automatic assessment, even one that is only moderately successful for some measures, and no welfare assessment existed, the choice is obvious. There is a clear need for some form of automatic welfare assessment, as it's the only way to get some facilities to enact any true form of injury within stalls or other welfare assessments. As such, I would suggest the authors minimize some of the discussion points, and focus on the following three points that could illustrate the importance of this study:
1. Cost: How much time, effort, and money would go into installing a system like this across an entire dairy farm? What about other settings? Is it applicable outside of just dairy farms? If the authors want to make their system applicable, then they should focus not just on the accuracy of their system, but how easy it would be both energetically and monetarily to implement.
2. Other methods: Are there other forms of automatic monitoring that could be implemented? How would this compare? The authors mentioned the GAITFour system in their intro; is this relevant, and why or why not? Noldus makes a video monitoring system that could potentially identify some of these contact behaviors automatically through video. Is that worth investigating?
3. What next?: Here, if the authors want to introduce how to improve their system, this could be mentioned, as well as follow-up experiments. What do you need to test next to show that you improved the ability of this system to assess contacts? Is that cost-efficient?
I think the simple point is that much of the Discussion as it stands could be reduced, with the majority of the results summarized into a couple of paragraphs. The authors could then focus on the three points noted above, which would actively argue for why automatic assessments of welfare are the next step in welfare assessment for dairy cows. Any other points in the Discussion section could be rolled into those three points, but I believe the majority of the focus should be on why and how to get more automatic welfare assessments enacted.
Author Response
Comments and Suggestions for Authors
Reviewer 2:
R2: This is a straightforward but important paper, as it details a means for automatically assessing the welfare of dairy cows. This is also potentially applicable to other settings, where a mechanized, automatic assessment of welfare could be extremely useful. The difficulties, as the authors note, is that the ability to accurately assess some of the contacts were only moderately correlated. While the authors spend some time in their discussion suggesting ways to improve these automatic assessments, it still remains at least a partial issue.
AU: Thank you for your review. The limitations of the CM system include its inaccuracy for 1) recording the exact number of contacts made by cows with the stall partitions (L299-300) and for 2) ranking cows by degree of contact with the neck rail (L255-257). As such, the authors only claim that the CM system can be used to accurately rank cows based on their frequency of contact with the stall dividers (L361-363). The validated component of the system holds true without any of the suggested improvements (L363-365).
R2: Regardless, if the choice between some form of automatic assessment, even one that is only moderately successful for some measures, and no welfare assessment existed, the choice is obvious. There is a clear need for some form of automatic welfare assessment, as it's the only way to get some facilities to enact any true form of injury within stalls or other welfare assessments. As such, I would suggest the authors minimize some of the discussion points, and focus on the following three points that could illustrate the importance of this study:
AU: The main objective of this research study was to validate the ability of the CM system to monitor cow contact with the stall dividers and neck rail when compared to video observation (L25-26). As such, the discussion was focused around providing support and limitations around the accuracy of the system from a validation perspective. Information relating to applicability in other contexts may be useful, but the system would require further validation outside of a tie-stall environment to ensure accuracy of the results (L271-277; L363-365). To support direct application of the validated CM system, information has been added to the discussion section according to the topics you recommended (L263-271; L276-277). However, extension of these topics to other contexts will not be considered, since it would warrant a separate validation study to ensure system accuracy.
R2: 1. Cost: How much time, effort, and money would go into installing a system like this across an entire dairy farm? What about other settings? Is it applicable outside of just dairy farms? If the authors want to make their system applicable, then they should focus not just on the accuracy of their system, but how easy it would be both energetically and monetarily to implement.
AU: Specifics on cost, time, and labour have been added to the discussion section (L263-271). This additional information should assist readers in deciding whether the system is feasible for application to their specific dairy farm (L263-265). The cost, time, and labour of applying the CM system to other settings cannot be commented on. To apply the CM system to settings other than a dairy farm, the system would likely need to be adapted to the new environment. The cost of modifications would depend on what changes had been made, in addition to the cost of running a validation study to determine system accuracy. This necessity for further modification and validation has been discussed with respect to application in a free-stall housing system (L271-277).
R2: 2. Other methods: Are there other forms of automatic monitoring that could be implemented? How would this compare? The authors mentioned the GAITFour system in their intro; is this relevant, and why or why not? Noldus makes a video monitoring system that could potentially identify some of these contact behaviors automatically through video. Is that worth investigating?
AU: Additional forms of automatic monitoring would require separate a research validation and lack relevance to the current study objective (L25-26). To the author’s knowledge, there are no other validated systems that can automatically monitor cow contact against the stall partitions to any extent (L360-361). While Noldus develops software to monitor animal activity, the technology has not been validated for use with dairy cows and does not claim to measure contact with the stall partitions. In addition, the GAITfour system was included in the introduction only as an example of automated behavioral monitoring and has only been tested as a walkway system (L62-64). Adapting these existing technologies for monitoring of cow contact with the stall partitions would be outside the scope of this study, and any attempt to do so would not be based on real data.
R2: 3. What next?: Here, if the authors want to introduce how to improve their system, this could be mentioned, as well as follow-up experiments. What do you need to test next to show that you improved the ability of this system to assess contacts? Is that cost-efficient?
AU: Any modifications to the CM system with the intention of improving its accuracy would necessitate a new validation study be conducted, in a similar format to the current study (L276-277). The cost efficiency of those modifications would depend on the specific changes made. It would likely be best to run a small pilot study on potential changes based on the limitations identified in the discussion section (L299-300; L255-257). Without data on the accuracy of potential improvements, we cannot recommend that they be implemented. Improvements have been suggested in the discussion section based on the limitations of the system (L271-276; L309-317). In terms of what comes next, the discussion section also includes future research questions and applications of the CM system (L258-263; L338-342; L353-358).
R2: I think the simple point is that much of the Discussion as it stands could be reduced, with the majority of the results summarized into a couple of paragraphs. The authors could then focus on the three points noted above, which would actively argue for why automatic assessments of welfare are the next step in welfare assessment for dairy cows. Any other points in the Discussion section could be rolled into those three points, but I believe the majority of the focus should be on why and how to get more automatic welfare assessments enacted.
AU: The discussion section as it pertains to the validation of the CM system for accuracy in monitoring cow contact against the stall partitions has not been altered (L25-26). Revisions related to the three new sections on system applicability and improvement have been added in (L263-271; L276-277). We agree that these points have improved the discussion section, by adding information on the feasibility of the system which can be compared to non-automated forms of data collection (i.e. visual observation). Thank you for your time and feedback.

Reviewer 3 Report
Comments to paper
Title: Validation of contact mats to monitor dairy cow contact with stall partitions
General comments:
This manuscript is well-written and has novelty and originality. The data presented are original and helpful to be used but my main concern is not having additional supportive data as it is necessary for the RESEARCH article. Do the authors have some supplementary data to be added to this manuscript either in the results or as a supplementary file (table or figure)? If not, I don’t know if “Animals” have regulations for “Short communications” papers which is advisable in this regard.
Simple summary:
Need one line definition of CM system for non-expert readers.
Introduction
If the author claim that “With the transition of dairy cow husbandry to more intensive systems, the welfare concerns associated with indoor housing have become a major research focus”; then why not using more updated references? Except for one belonging to 2017, other references are somehow old. Please add more updated references, if any, to the Introduction. At least two more references should address the issue and between citations of 2016-2019. However, I understand there is a lack of works on this new subject.
Line 91: “ad libitum” should be italic.
Results
Prefer additional supplementary data if any.
Discussion
It has written very well.
Author Response
Comments and Suggestions for Authors
Reviewer 3:
R3: General comments - This manuscript is well-written and has novelty and originality. The data presented are original and helpful to be used but my main concern is not having additional supportive data as it is necessary for the RESEARCH article. Do the authors have some supplementary data to be added to this manuscript either in the results or as a supplementary file (table or figure)? If not, I don’t know if “Animals” have regulations for “Short communications” papers which is advisable in this regard.
AU: In accordance with the requirement by Animals to include research data and maintain transparency, a Supplementary Materials section has been added (L369). Table S1 (L370-377) and Table S2 (L378-382) show the average number of contacts per minute made by each cow (n = 11) with the dividers and tie-rail, respectively, as recorded by the CM system compared to video observation (Rail Contact, CM Contact, Effective CM Contact). This is the data that was used to run the correlational analysis and validate the CM system. It has not been added to the results directly, since it is the data used to create Figure 2 (L318-321) and so would be redundant to include in the main text of the manuscript.
R3: Simple summary - Need one-line definition of CM system for non-expert readers.
AU: A one-line definition of a CM has been added on L11-12, to describe that a contact mat (CM) is a metal band that produces an electrical signal in response to a contact force.
R3: Introduction - If the author claims that “With the transition of dairy cow husbandry to more intensive systems, the welfare concerns associated with indoor housing have become a major research focus”; then why not using more updated references? Except for one belonging to 2017, other references are somehow old. Please add more updated references, if any, to the Introduction. At least two more references should address the issue and between citations of 2016-2019. However, I understand there is a lack of works on this new subject.
AU: Two additional references have been added to the introduction section to support the point that stall configurations which impede movement are associated with a higher prevalence of injuries in dairy cows (Bouffard et al., 2017; Nash et al., 2016) (L51-52).
R3: Line 91: “ad libitum” should be italic.
AU: The term “ad libitum” was italicized on L93.
R3: Results - Prefer additional supplementary data if any.
AU: As mentioned, a Supplementary Materials section has been added (L369) with Table S1 and S2 showing the research data that was included in the correlational analysis for CM system validation.
R3: Discussion - It was written very well.
AU: Thank you for your feedback.
